# A Multimodal Late Fusion Framework for Physiological Sensor and Audio-Signal-Based Stress Detection: An Experimental Study and Public Dataset

**Vasileios-Rafail Xefteris** [1,*], **Monica Dominguez** [2], **Jens Grivolla** [2], **Athina Tsanousa** [1], **Francesco Zaffanela** [3], **Martina Monego** [3], **Spyridon Symeonidis** [1], **Sotiris Diplaris** [1], **Leo Wanner** [2,4], **Stefanos Vrochidis** [1] **and Ioannis Kompatsiaris** [1]

[1]   Centre for Research and Technology Hellas, Information Technologies Institute, 6th Km Charilaou-Thermi, 57001 Thermi, Greece; atsan@iti.gr (A.T.); spyridons@iti.gr (S.S.); diplaris@iti.gr (S.D.); stefanos@iti.gr (S.V.); ikom@iti.gr (I.K.)

[2]   Department of Information and Communication Technologies, Pompeu Fabra University, Roc Boronat, 138, 08018 Barcelona, Spain; monica.dominguez@upf.edu (M.D.); jens.grivolla@upf.edu (J.G.); leo.wanner@upf.edu (L.W.)

[3]   Autorita di Bacino Distrettuale delle Alpi Orientali, Cl. Seconda del Cristo, 4314, 30121 Venice, Italy; francesco.zaffanella@distrettoalpiorientali.it (F.Z.); martina.monego@distrettoalpiorientali.it (M.M.)

[4]   Catalan Institute for Research and Advanced Studies, Passeig Lluís Companys, 23, 08010 Barcelona, Spain

*   Correspondence: vxefteris@iti.gr

**Abstract:** Stress can be considered a mental/physiological reaction in conditions of high discomfort and challenging situations. The levels of stress can be reflected in both the physiological responses and speech signals of a person. Therefore the study of the fusion of the two modalities is of great interest. For this cause, public datasets are necessary so that the different proposed solutions can be comparable. In this work, a publicly available multimodal dataset for stress detection is introduced, including physiological signals and speech cues data. The physiological signals include electrocardiograph (ECG), respiration (RSP), and inertial measurement unit (IMU) sensors equipped in a smart vest. A data collection protocol was introduced to receive physiological and audio data based on alterations between well-known stressors and relaxation moments. Five subjects participated in the data collection, where both their physiological and audio signals were recorded by utilizing the developed smart vest and audio recording application. In addition, an analysis of the data and a decision-level fusion scheme is proposed. The analysis of physiological signals includes a massive feature extraction along with various fusion and feature selection methods. The audio analysis comprises a state-of-the-art feature extraction fed to a classifier to predict stress levels. Results from the analysis of audio and physiological signals are fused at a decision level for the final stress level detection, utilizing a machine learning algorithm. The whole framework was also tested in a real-life pilot scenario of disaster management, where users were acting as first responders while their stress was monitored in real time.

**Keywords:** stress detection; multimodal fusion; physiological signals; audio analysis

## 1. Introduction

The physiological reaction of a person, when exposed to challenges of high discomfort and difficulty (also referred to as "stressor" [1]), is defined as stress. Stress can influence a person's performance and mental lucidity, thus making it one of the most important aspects of disaster management applications. Apart from the short-term effects of stress, exposure to stress for long periods of time can have serious effects on the health of a person. Some of the most common health problems induced by chronic stress include post-traumatic stress disorder (PTSD) and major depressive disorder [2], or other physical health problems, such as sleep disturbances and musculoskeletal problems [3]. Thus, the monitoring of

stress levels of first responders, who are often exposed to highly stressful situations, can be essential both for their performance and their health.

Since stress is a physiological reaction, monitoring physiological signals can offer valuable information for accurate stress detection. The recent developments in Internet of Things (IoT) devices have led to increasing research in the field of automatic stress detection by monitoring physiological signals. Among the most common physiological sensors exploited for stress detection are the galvanic skin response [4] (GSR), heart rate [5] (HR), respiration signals [6] (RSP), electrocardiograph signals [7] (ECG), or even studying brain dynamics with sensors such as electroencephalography [8] (EEG). In other cases, subjects' movements may also reflect their stress levels. Thus, monitoring muscle activity through electromyography (EMG) sensors or kinematic data through inertial measurement unit (IMU) sensors has also been proposed as a solution for automatic stress detection [9].

Nevertheless, stress can also be reflected in human behavior as a result of abnormal physiological functions, e.g., alteration in heart rate and heart rate variability, breathing patterns, or even muscle tension of the vocal cords [10]. During stress, speech patterns can be influenced, leading to alterations in speech jitter, energy in certain frequency bands, or shift in the fundamental speech frequency [10,11]. Such physical alterations of the speech signal can be captured by digital recording devices. The recorded speech signal is processed, and specific acoustic features are extracted for analysis and prediction using machine learning techniques. The development of devices such as smartphones, which are equipped with microphones, allows for the monitoring of users' speech and real-time processing for accurate and effective speech-based stress detection.

The unique characteristics of physiological signals and speech patterns can complement the task of automated real-time and accurate stress level detection. Therefore, the study of fusing such data for the cause of stress detection is of great interest. Nevertheless, this fusion scheme has not been adequately studied in the literature. One of the main reasons for this knowledge gap is the lack of publicly available datasets that can allow researchers to develop and compare multimodal fusion models for stress detection based on such data types. Our work aims at introducing such a multimodal dataset for automatic stress detection including physiological signals and speech cues. A data collection protocol was designed to induce different levels of stress to the users in a controlled environment, while their physiological and speech data were collected. The protocol is based on a sequence of well-known and frequently applied stressors, both psychological and physical. For the collection of physiological signals, a smart vest equipped with ECG, RSP and IMU sensors was designed, and users were asked to wear it during the whole duration of the experiments. For the audio monitoring, an application was developed allowing for recording and posting audio recordings in real time. The ground truth values were self-annotated by the participants after the completion of each stressor in a continuous-valued manner. The same configuration was also used in a real-life scenario of disaster management, where subjects are serving as first responders during a hypothetical flood emergency in Vicenza, Italy. We also present a proposed real-time analysis of the collected data based on state-of-the-art physiological signals and audio analysis methods and a decision-level fusion approach, where the laboratory experiment data were used for training and the real-life pilot data for evaluation. The main contributions of our paper include the following:

- We provide a multimodal dataset including physiological signals and speech cues for stress detection. To our knowledge, this is the first dataset providing such data types for the purpose of stress detection.
- Our dataset also includes data from a real-life disaster management scenario.
- We describe the decision-level fusion framework adopted that includes state-of-the-art machine learning-based methods for the analysis of each modality.

The rest of the paper is organized as follows: In Section 2, a review of the state of the art in the field of stress detection is presented, focusing on publicly available datasets as well as analysis methods for physiological data, audio information, and their fusion.

In Section 3, the methods for the data collection and the proposed analysis are described in detail, followed by the results in Section 4. Finally, Section 5 concludes the paper.

## 2. Related Work

In this section, the current state of the art in the field of stress detection is reviewed, investigating both the publicly available datasets in the literature as well as analysis methods for stress detection based on physiological signals, auditory cues, and their fusion.

### *2.1. Analysis Methods*

#### 2.1.1. Physiological Signal-Based Stress Detection

Considering the physiological nature of stress, there has been increased interest in the applications of monitoring physiological signals for automated stress detection.

In the work by Gil-Martin et al. [12], the authors proposed a multimodal physiological signal solution for automated stress detection based on a convolutional neural network (CNN) deep learning architecture and several signal processing techniques. For this cause, they used the publicly available WESAD [13] dataset. Their results revealed that the combination of all the signals using a Fourier transform and cube root processing techniques can improve the overall performance, and when applying a constant Q transform over the previous processing, the input data shape can be significantly reduced while maintaining performance. They achieved an accuracy of 96.6% when dealing only with two classes (stress–no stress). The same dataset has been extensively used in combination with machine learning [14,15] or deep learning [16,17] algorithms in the context of the stress detection problem.

#### 2.1.2. Audio-Based Stress Detection

Detecting stress using audio-signal-based techniques requires two main processes: (i) acoustic feature processing and extraction and (ii) prediction (or classification) of the estimated level of stress. At a high level, these tasks can be described as audio-signal processing, data processing, and statistical inference via machine learning algorithms.

In the MuSE challenge [18], several feature sets from several modalities (i.e., heart rate, face movements, etc.) are used to predict stress annotations. The feature set that outperforms the task of stress detection is precisely the acoustic feature set. Specifically, they use the open-source software OpenSMILE 3.0 [19] with a predefined set of 88 acoustic features, known as the eGeMAPs feature set [20]. Prediction of continuous values of stress at 500 ms windows is performed, and an overall accuracy of 0.44 on the MuSE challenge's test set is reported. Such results can be used as a benchmark that highlights the non-trivial nature of the task at hand for audio-based stress detection.

#### 2.1.3. Fusion of Physiological Signals and Vocal Cues

Since stress detection can be addressed using either physiological signals or speech information, the combination of these two modalities could be proved to be valuable in the task of stress detection using wearables. Nevertheless, this field of research has not been adequately studied. In the work of Kim et al. [21], the authors proposed a combination of speech cues and physiological signals including ECG, EMG, RSP, and GSR for emotion recognition during a virtual quiz simulation. Their results indicate that fusing speech and physiological signals at the feature level using a feature selection method is the best-performing method for emotion recognition. In their follow-up work [22], they proposed a similar approach for emotion based on short-term observations, contrary to their previous work. In both studies, they used the same dataset of the virtual quiz wizard. Their results again revealed that feature-level fusion with feature selection outperforms other fusion methods as well as unimodal methods. Kurniawan et al. [23] proposed a scheme for fusing GSR and speech signals for stress detection induced by a mental workload of varying difficulty. The authors tested two different fusion approaches; a feature-level fusion scheme

and a decision-level fusion scheme. Their results indicate that the decision-level fusion using a support vector machine (SVM) classifier achieved an accuracy score of 92.47%.

*2.2. Public Datasets*

2.2.1. WESAD Dataset

The WESAD dataset [13] is a physiological signals dataset for stress and amusement detection using wearables that record physiological and motion data. The signal acquisition was performed using two wearable devices; a chest-worn device equipped with a three-axis accelerometer, ECG, electrodermal activity (EDA), electromyography (EMG), RSP sensor, and temperature (TEMP) sensors and a wrist-worn device equipped with a three-axis accelerometer, EDA, blood volume pulse (BVP), and TEMP sensors. The experiment was based on alterations between amusement states, induced by funny videos, relaxation and meditation, and stressful situations, through the Trier Social Stress Test stressor. The ground truth was acquired from self-reports from the participants.

2.2.2. SWELL-KW

The SWELL-KW dataset [24] is a multimodal dataset for stress detection of workers during typical knowledge work. The dataset was collected during an experiment, where 25 participants had to perform typical office knowledge work and time pressure and email interruptions were used as the stressors. During the experiment, behavioral data, in terms of facial expression from a camera, body postures from a Kinect 3D and computer logging, and physiological signals from heart rate and skin conductance sensors were monitored. The ground truth assessment was performed with validated questionnaires regarding the task load, mental effort, emotion, and perceived stress.

2.2.3. DRIVE-DB

The DRIVE-DB dataset [25] is a multimodal dataset for detecting driver stress levels from nine subjects. The dataset includes video data from the drivers' heads to detect head movement and also physiological signals from RSP, EMG, ECG, HR, and GSR sensors captured in an ambulatory environment.

2.2.4. Comparison with Our Dataset

Even though other datasets exist, including physiological signals and behavioral data, in terms of facial expression in the SWELL-KW dataset or driver's head video in the DRIVE-DB dataset, no dataset including physiological signals and auditory cues is available. Since both modalities have been proven to be valuable in the task of stress detection, the study of their possible complementary nature is of great interest. Compared to face front videos, audio input as behavioral data has the benefit of not being prone to positioning and occlusion problems, therefore being easier to implement in real-life applications, such as the disaster management application described in this paper.

**3. Methods**

*3.1. Stress Induction Protocol*

The stress induction protocol was designed to differentiate between various levels of stress and calmness in a controlled way. All subjects had to perform specific tasks, which included well-known stressors and actions for relaxation. These tasks can be divided into psychological and physiological tasks, according to the type of stress they are related to, and they are the following:

- Psychological:
    - The Stroop test (Figure 1a). It is a well-known stressor [26]. In the Stroop test, certain words of color names are written in a different color than the one they describe. The user, after seeing each slide of words for a short period of time, is asked to describe the font color of each word.

&ndash; The descending subtraction test (Figure 1b). It is also a commonly used stress induction test [27], where the user is asked to begin counting backward from a certain number, subtracting each time another certain number. In the context of the training data collection experiment, the users were asked to begin with the number 1324, subtracting 17, until 17. If the users make a mistake, they must start over.

&ndash; Explain a stressful situation in your life.

&ndash; Explain how the day has been.

&ndash; Listen to relaxing music. The two later tasks are used to monitor situations of calmness.

- Physiological:

&ndash; Place a hand in cold water (2 °C) for two minutes, make a pause, and then place it again (Figure 1c).

&ndash; Ascend and descend four levels of stairs (Figure 1d).

&ndash; Tie and untie shoes after exercise (Figure 1e).

After each task, the subjects were asked to report their stress level as a continuous value in the range of 0–100.

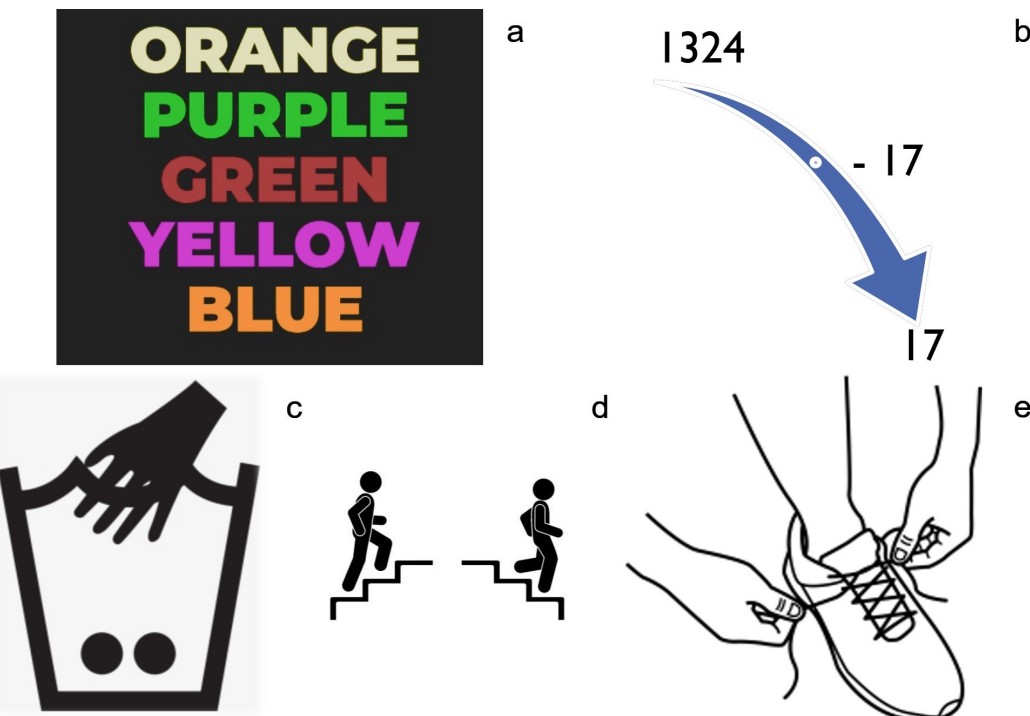

**Figure 1.** The psychological (upper half of the figure) and physiological (lower half of the figure) stressors that users had to perform. (**a**) The Stroop test, (**b**) the descending subtraction test, (**c**) placing hand in cold water, (**d**) ascend and descend stairs, (**e**) tie and untie shoes.

### 3.2. Data Acquisition

Most of the domain-specific data used for training and evaluating the sensor- and audio-based stress detection modules were provided or generated by the Alto-Adriatico Water Authority/Autorità di bacino distrettuale delle Alpi orientali (AAWA). An ethics advisory board was monitoring all the research for ethical and legal compliance, and all participants were asked to sign a consent form prior to the experiments. All methods were carried out in accordance with relevant guidelines and regulations. All experimental protocols were approved by the competent ethics authority. Informed consent was obtained from all subjects and/or their legal guardian(s). The data acquisition techniques can be divided into physiological and audio data collection.

### 3.2.1. Physiological Data Collection

For the physiological data acquisition, a smart vest equipped with IMU, ECG, and RSP sensors, which can be seen in Figure 2, was used. In more detail, the smart vest has integrated textile sensors for the acquisition of ECG and RSP signals and a data logger for the transmission of data via Bluetooth 2.1. The data logger has an integrated IMU, including accelerometer, gyroscope, magnetometer, and quaternion sensors, which can be used to monitor the movements of the trunk. The data logger has also a developed software for extracting information from the monitored signals, such as heart rate from the ECG signal and breathing rate from the RSP signal. A detailed description of the monitored parameters, along with their sampling rates and units, is presented in Table 1.

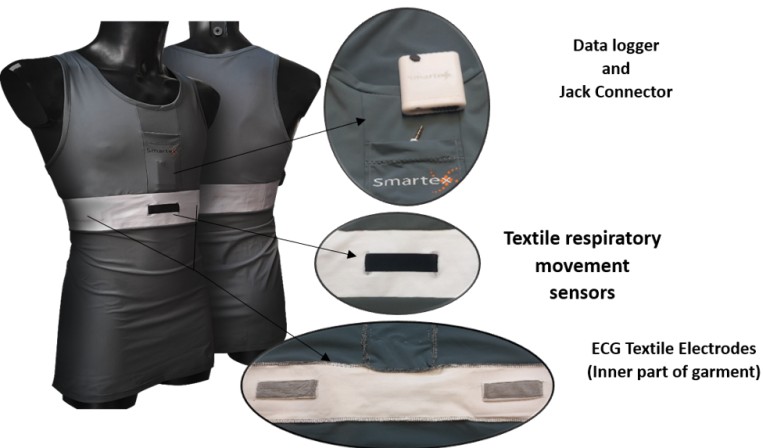

**Figure 2.** Wearable smart vest architecture.

**Table 1.** Smart vest recorded parameters.

| Recorded Parameter | Description | Values (per 1 Unit Metric) | Sampling Rate |
| --- | --- | --- | --- |
| ECG Value | Electric signal measuring the ECG | 0.8 mV | 250 Hz |
| ECG quality Value | ECG signal quality | 0–255 (0 = poor, 255 = excellent) | 5 Hz |
| ECGHR Value | Heart rate | Beats/minute | 5 Hz |
| ECGRR Value | R-R intervals | number of samples between R-R peaks | 5 Hz |
| ECGHRV Value | Heart rate variability | ms | 60 Hz |
| AccX-Y-Z Value | Acceleration in X-Y-Z axes | $0.9710^{-3}$ g | 25 Hz |
| GyroX-Y-Z Value | Angular velocity in X-Y-Z axes | 0.122°/s | 25 Hz |
| MagX-Y-Z Value | Magnetic field in X-Y-Z axes | 0.6 µT | 25 Hz |
| Q0-Q1-Q2-Q3 Value | Quaternions from main electronic device (Q0, Q1, Q2, Q3 components) | Q14 format | 25 Hz |
| RespPiezo Value | Electric signal measuring the chest pressure on the piezoelectric point | 0.8 mV | 25 Hz |
| BR Value | Breathing rate | Breaths/minute | 5 Hz |
| BA Value | Breathing amplitude | logic levels | 15 Hz |

### 3.2.2. Audio Data Collection

For training the audio-based stress detector, material from AAWA had been provided in the domain of citizens' phone calls reporting emergencies. The dialogues were simulated but closer to real-world emergency management contexts than any other material previously used for developing the stress module. A total of 25 phone dialogues between citizens and operators were provided, including mostly male voices usually performing different roles (as citizen and as operator). The task of annotating this material to gather training data for the stress module consisted of several steps: (i) choosing an annotation tool, (ii) developing annotation guidelines, (iii) preparing the material for an-

notation, (iv) following up the annotator's completion of the task, and (v) processing the material for training. We chose as our annotation tool the open source software NOVA (https://github.com/hcmlab/nova) [28] due to its user-friendly interface and compatibility with Windows OS. NOVA allows frame-wise labelling for a precise coding experience and value-continuous annotations for labelling, e.g., emotions or social attitudes, including perception of stress in voice. The interface is customizable and allows loading and labelling data of multiple persons. The resulting continuous annotation can be exported as a csv file with timestamps.

Step-by-step annotation guidelines were provided to user partners from AAWA who kindly helped out in the annotation task, including a demonstration video on how to use the NOVA software. In order to have a minimum amount of material for training a model, three annotations from different people are needed. The 20 dialogues were segmented into dialogue turns to isolate each speaker utterance for the annotation task. A total of 262 audio files were used for the annotation of stress. Three rounds of annotations were carried out, and a total of 11 annotators took part in the process to split the amount of material and thus efficiently distribute the task. Thus, the minimum required amount of three annotations for each audio file were obtained.

For testing the multimodal stress detection, audio recordings were gathered along with physiological signals in the experiment described in the "Stress Induction Protocol" section above.

### 3.3. Participants

During the data collection experiments, a total of 5 subjects participated, of which 2 were female. Their age ranged from 30 to 50, with a mean age of $37.6 \pm 7.6$. The whole duration of the experiments was about 32 min and 40 s for each participant, and during the whole duration of the experiments, they were wearing the developed smart vest and they were recorded through the developed smartphone application. All participants signed the necessary consent forms. Along with the subjects, a supervisor was present during the whole experiment to ensure the correct execution of the experimental process. During the disaster management pilot, a total of 11 subjects participated, operating as first responders, while wearing the developed smart vest and also sending audio reports through the developed application.

### 3.4. Data Analysis

3.4.1. Physiological Data Analysis

The analysis of physiological data consists of two steps; preprocessing and feature extraction. The whole physiological signals data analysis was performed in a sliding window of 60 s with 50% overlap. Only the monitored physiological signals were used in this analysis; those being the ECG data (ECG Value), RSP data (RespPiezo Value), and IMU data (AccX-Y-Z Value, GyroX-Y-Z Value, MagX-Y-Z Value, and Q0-Q1-Q2-Q3 Value). The parameters extracted from the physiological signals using the data logger software (ECG quality Value, ECGHR Value, ECGRR Value, ECGHRV Value, BR Value, and BA Value, see Table 1) were not used in this analysis. For the preprocessing of the physiological signals, multiplication with simple weights has been performed as a first step in order to convert the units of the signals into the metric system (see Table 1). After the first step of preprocessing the data, each one of the modalities was processed differently to further preprocess them and extract valuable features from them.

Regarding the IMU sensors, the only sensor that was further preprocessed was the quaternion, in order to extract roll, pitch, and yaw angles from it. After the preprocessing is completed, the feature extraction includes 10 time-domain features; namely mean, median, standard deviation, variance, maximum value, minimum value, interquartile range, skewness, kurtosis, entropy, and 6 frequency-domain features; namely energy, and 5 dominant frequencies. These features were computed for each axis of the sensors integrated in the IMU for a total of 192 features.

For the ECG analysis, before extracting features, two different filters were applied; a 5th-order Butterworth high-pass filter at 0.5 Hz, in order to remove baseline drift, and a 50 Hz notch filter to remove powerline frequency noise. Finally, the preprocessing of the ECG signal also includes a peak detection technique in order to extract information regarding the QRS complex peaks and the R-R intervals, which are the physiological phenomenon of variation in the time interval between heartbeats. The R-R intervals can be used to extract heart rate information, such as heart rate variability (HRV). The extracted ECG features include time- and frequency-domain statistical features regarding the R-R intervals and the HRV by using the hrv-analysis [29] and the neurokit [30] toolboxes, resulting in a total of 94 ECG features.

The preprocess of the RSP signal was performed following the methods proposed by Khodadad et al. [31], leading to the identification of the inhalation and exhalation peaks. After the preprocessing of the RSP signals, time- and frequency-domain statistical features were extracted regarding the breathing rate, respiratory rate variability, and breath-to-breath intervals. The RSP signal analysis was also performed using the neurokit [30] toolbox, resulting in a total of 28 RSP features.

After all the features were extracted, four different machine learning algorithms were tested; namely random forest (RF), k nearest neighbor (kNN), SVM with linear basis, and eXtreme gradient boosting tree (XGB), using 10-fold cross-validation. Different fusion and feature selection methods were tested, which are described in detail in [32]. For the ground truth values, each window received the self-reported stress level value that refers to the task each user was performing during the certain window. The stress-level values were normalized in the 0–1 range.

### 3.4.2. Audio Data Analysis

Post-processing of annotated material was needed. We processed inconsistent file naming and computed the mean average score for each audio both as continuous values of stress in each audio file (at 40 milliseconds frames) and as one overall score per audio file. In addition, the material was split into training and testing sets for machine learning experiments and validation.

There are only 262 audio files (accounting for speaker turns in a total of 20 citizen–operator dialogues). Some speakers perform the roles of both citizen and operator. There is only one female voice (acting as operator in 2 dialogues and as citizen in 1). Annotations are normally distributed with a skewness towards the left, which means there are considerably more annotations in quartile 1 (that is, stress level around 0.3 on a scale from 0 to 1), as seen in Figure 3. Results from this analysis imply that the normal distribution of stress should also be considered around level 0.3 in the output of the model.

For feature extraction, the OpenSMILE API [19] was used both locally and by calling the available Python library in two different scenarios: (i) using the predefined configuration extracting a feature set of 88 acoustic features from the whole audio file, known as eGeMAPs feature set [20], and (ii) using the predefined configuration extracting 10 low-level-descriptor (LLD) features at 25 ms windows with 10 ms steps. Two sets of acoustic features were derived and post-processed for classification experiments to match the ground truth values. These features were then used to train different classifiers using the WEKA framework [33]. The best results were obtained with a Gaussian process [34] regression model.

Alternatively, features can also be calculated on longer audio segments instead of short windows. Having short overlapping windows is useful to produce a continuous estimation of stress levels over time, and thus analyze the evolution of stress levels over a longer time period (with a corresponding continuous audio recording). However, when audio is only recorded intermittently, as was the case in the pilot tests performed in the project (see section "Pilot Results" below), it can be useful to treat each recording (up to a few seconds in length) as a single unit with a single predicted stress level.

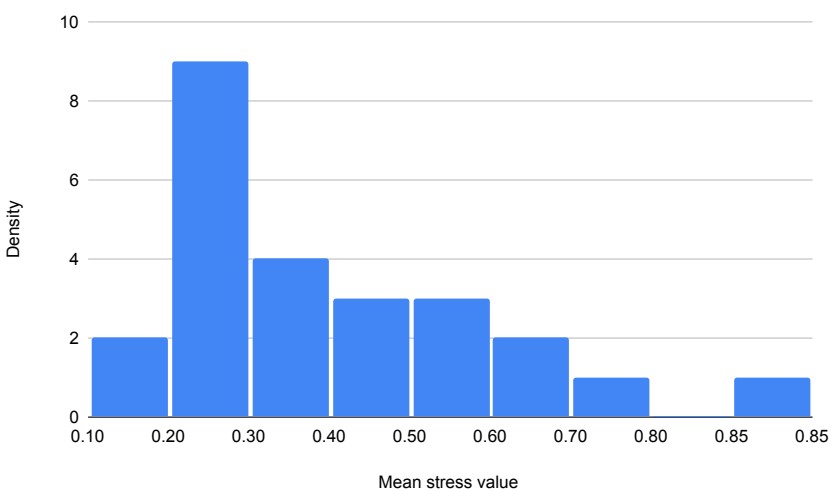

**Figure 3.** Distribution of manually annotated stress levels for audio-based detection.

### 3.5. Fusion of Physiological and Audio Stress Levels

After both the physiological sensor- and audio-based stress level detection modules were trained, a decision-level multimodal fusion was performed in order to further improve the performance of the system. The results from both the sensor- and audio-based models were synchronized, and then different methods were tested for decision-level fusion. More specifically, the performance of five different machine learning algorithms was tested; namely kNN, RF, SVM with radial basis function, SVM with linear basis function, and XGB. For the kNN algorithm, the number of neighbors was set to k = 5. The number of trees in the RF model was set to 100. For both the SVM linear and SVM radial, the regularization parameter was set to C = 1. Finally, the number of trees for the XGB algorithm was set to 100. The results from both modalities were used as inputs for the fusion models as a concatenated feature set. The performance of a GA-based weighted average method was also evaluated. For all of the different methods, a 10-fold cross-validation technique was incorporated and the MSE metric was used for the final evaluation.

## 4. Results

### 4.1. Training Results

#### 4.1.1. Physiological Sensor Results

The results of the different early and late fusion methods used for the analysis of physiological signals are presented in Table 2. From the table, it can be seen that the fusion method achieving the lower MSE score is the feature-level fusion of all modalities using the XGB machine learning algorithm; that being 0.0730. It is also worth noticing that the IMU outperforms the other modalities, which might indicate that the physiological stressors, which include specific body movements, might have a higher influence on the users' stress levels.

**Table 2.** MSE results of the different fusion techniques with all four different regressors.

|  | ECG | RSP | IMU | ECG + RSP | ECG + IMU | RSP + IMU | ECG + RSP + IMU | Late Mean | Late Median |
|---|---|---|---|---|---|---|---|---|---|
| **SVM** | 0.1709 | 0.1530 | 0.1305 | 0.1723 | 0.1306 | 0.1305 | 0.1305 | 0.1412 | 0.1363 |
| **kNN** | 0.1439 | 0.1553 | 0.1107 | 0.1285 | 0.1106 | 0.1106 | 0.1107 | 0.1170 | 0.1125 |
| **RF** | 0.1113 | 0.1280 | 0.0918 | 0.1073 | 0.0916 | 0.0871 | 0.0886 | 0.0984 | 0.1025 |
| **XGB** | 0.1237 | 0.1307 | 0.0844 | 0.1092 | 0.0835 | 0.0858 | 0.0730 | 0.0958 | 0.1006 |

In addition, three different feature selection methods were evaluated in terms of their MSE score. Since the best-performing combination of features includes all the modalities, the feature selection was performed on the combined feature set. The results of the different

feature selection methods are presented in Table 3. Of the different feature selection methods, the one having the best result is the GA-based feature selection. Again, the XGB machine learning algorithm was the one having the lower MSE score; that being 0.0567.

**Table 3.** MSE results of the different feature selection techniques with all four different regressors.

|  | RFE | PCA | GA |
|---|---|---|---|
| **SVM** | 0.1052 | 0.1201 | 0.1305 |
| **kNN** | 0.1023 | 0.1106 | 0.1106 |
| **RF** | 0.0790 | 0.1044 | 0.0742 |
| **XGB** | 0.0772 | 0.0953 | 0.0567 |

Following these results, an XGB machine learning algorithm was trained based on the subset of features selected using the GA-based feature selection method. This model was deployed for the real-time stress-level prediction in the real-life pilot scenario.

4.1.2. Audio-Based Results

A train–test split was performed, where 237 samples were chosen as a training set and the remaining 25 samples were chosen as the test set. Using a bagging machine learning model, the results showed that the use of the whole 88 eGeMAPs feature set achieved an MSE score of 0.01. These results of the 25 test samples are also presented in Figure 4. In the figure, the x-axis represents each one of the test samples and the y-axis the stress levels in the range of 0 to 1.

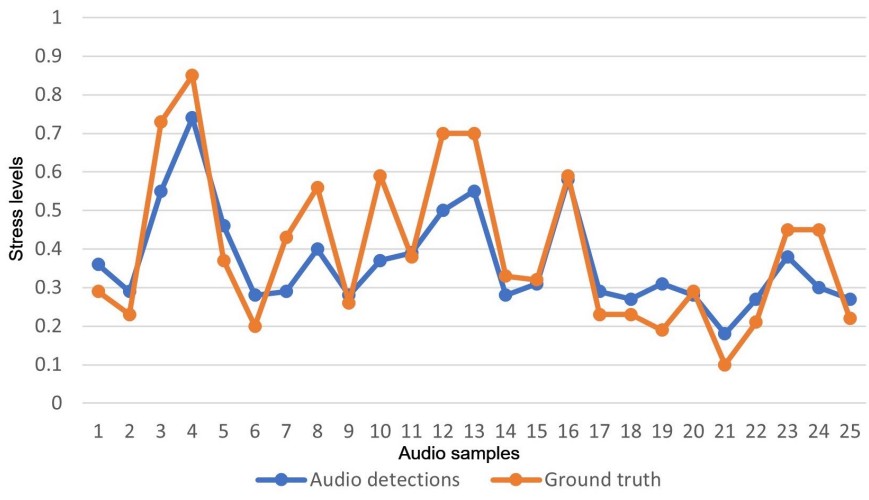

**Figure 4.** Results of audio-based stress-level detections versus ground truth values.

4.1.3. Fusion Results

For the multimodal fusion training and testing, both the sensor-based and audio-based modules extracted results from the data acquired during the experimental protocol described in the "Stress Induction Protocol" section above. The MSE results of the different machine learning algorithms and the GA-based weighted averaging method applied for the decision-level fusion of physiological signals and audio cues are presented in Figure 5. From the figure, it can be seen that the machine learning methods have a much better performance than the GA-based weighted averaging method. That might indicate that the machine learning algorithms reveal higher-level correlations than the GA-based weighted averaging technique. Out of the five different machine learning algorithms, the best performing is the SVM with a radial basis function, achieving an MSE score of 0.0062.

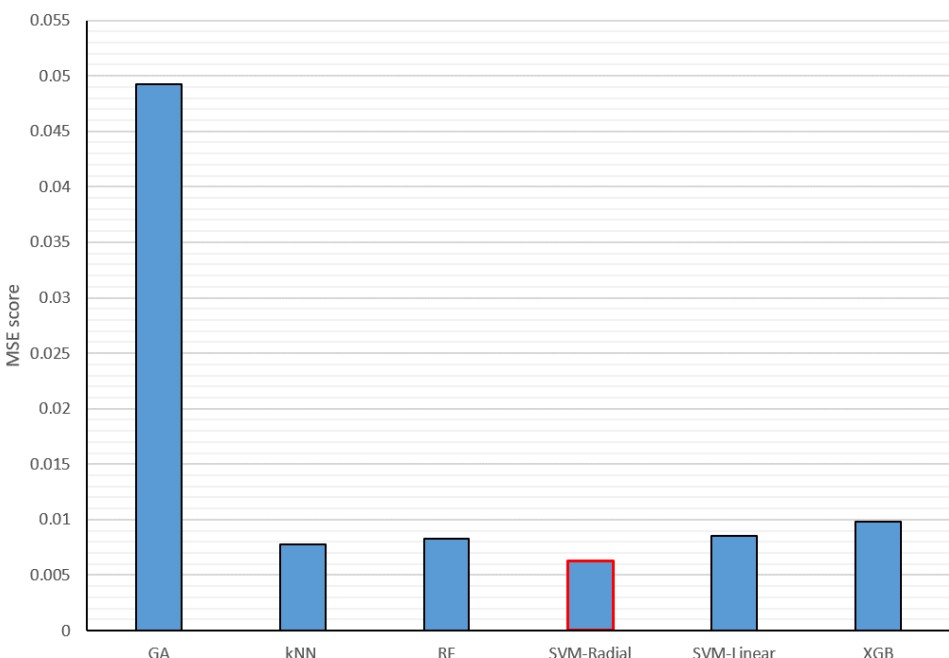

**Figure 5.** Physiological signals and audio decision-level fusion MSE results for different machine learning algorithms. The best-performing method is outlined in red.

*4.2. Pilot Results*

For the disaster management real-life pilot scenario, the best-performing models for each of the physiological-, audio-, and fusion-based stress level detection were trained and deployed. In order to acquire the necessary physiological data, the users were wearing the smart vest during the whole duration of the pilot. For the acquisition of audio data, a smartphone application was developed, allowing users to send voice messages when they decide to. The same application was also used for the stream of the physiological data to feed them to the sensor-based model for stress detection.

The overall workflow of the pilot scenario data analysis is presented in Figure 6. The workflow for the data analysis during the pilot is as follows:

- Physiological signals are continuously monitored using the smart vest.
- Physiological data are fed to the sensor-based stress-level detection module, which has the following operation:
    - Stack packages of data until one minute duration is reached. Since the smart vest produces 5 s long packages of data, 12 packages are stacked each time.
    - Features are extracted and selected as described in the previous sections.
    - Feed selected features to the trained XGB model for physiological sensor-based stress detection.
- If audio recordings are received, they are analyzed with the following operation:
    - Audio signals are optionally segmented (especially for longer recordings).
    - Feature extraction is performed following the procedure described in the "Audio Data Analysis" subsection.
    - Features are fed to the trained Gaussian process regression model for the audio-based stress-level detection.
- Sensor-based stress level and audio-based stress level are fed to the fusion SVM model for the fused stress-level detection.

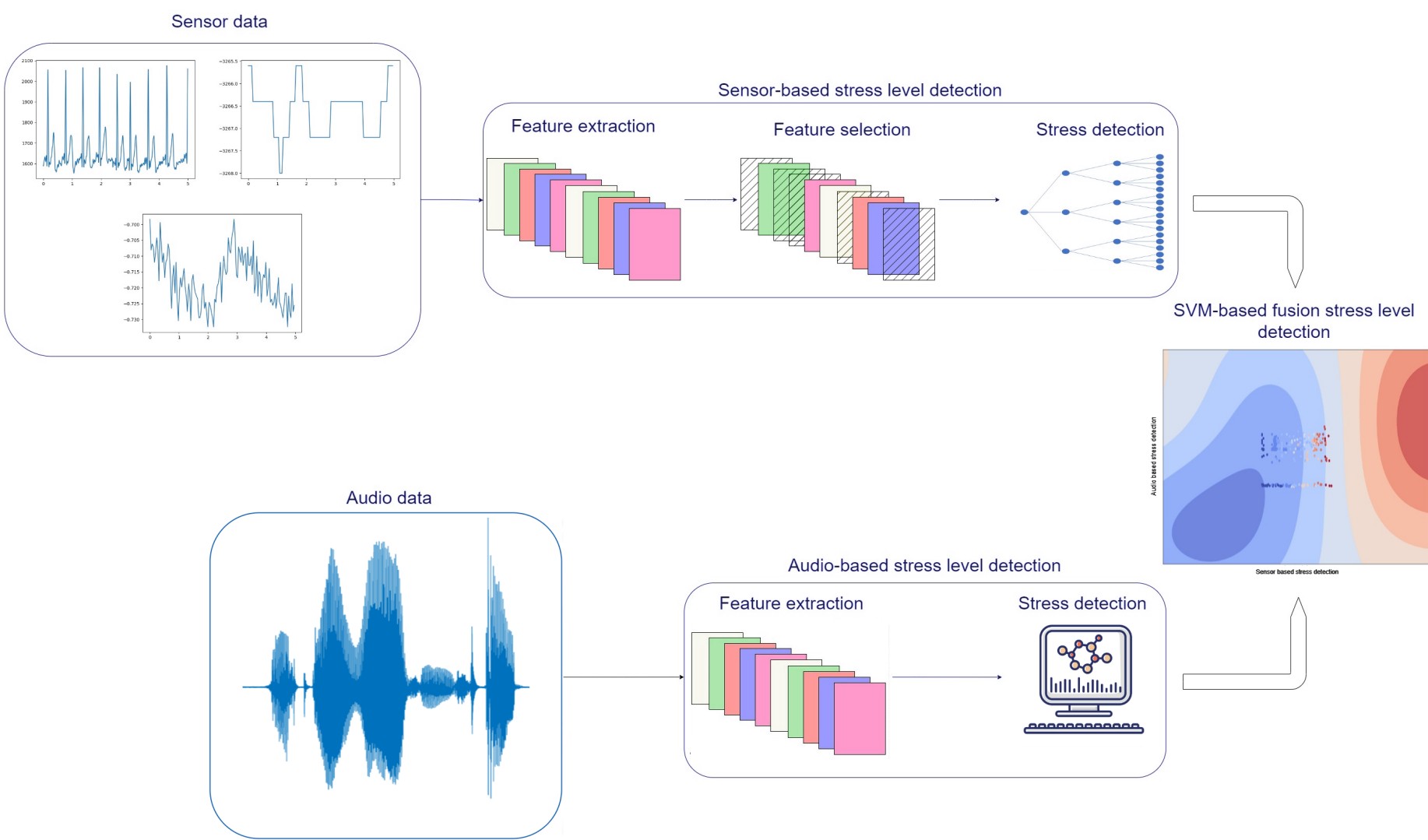

**Figure 6.** Graphical representation of the fusion stress-level detection pipeline during the pilots.

The results from the pilot data are presented in Figure 7. Each sub-figure depicts the results of 1 of the 11 subjects that participated in the pilots. The x-axis in each of the sub-figures represents time and the y-axis represents the stress level on a scale from 0 to 1. The results of Figure 7 indicate that across all subjects, the stress levels remained at a medium level throughout the whole pilot duration. This is a reasonable result since, during the pilot experiments, there was not any real stressor, such as a real flood simulation, that could induce high levels of stress.

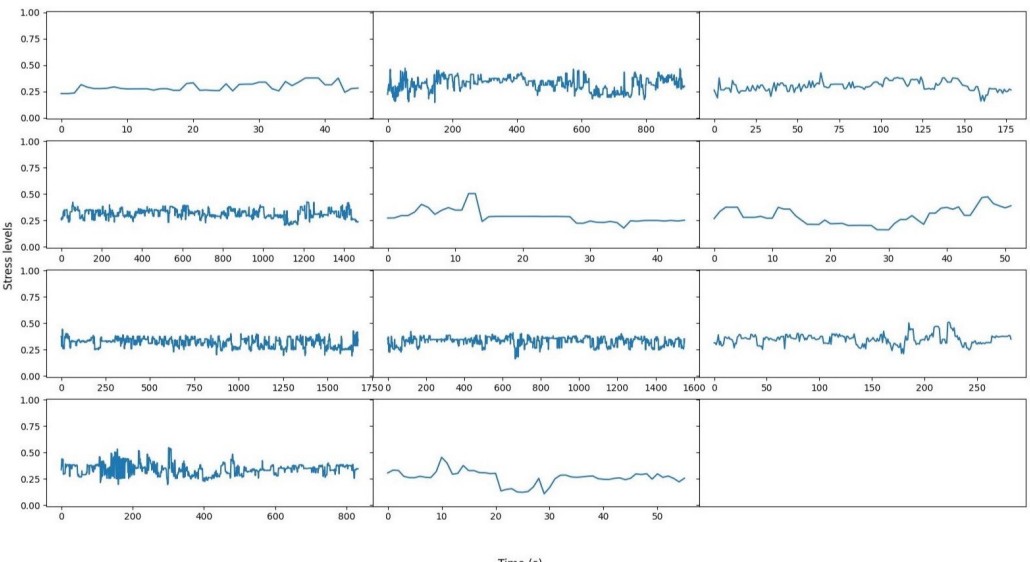

**Figure 7.** Fusion results of stress over time results from the pilot data. Each sub-figure represents the stress over time of one subject. The x-axis of each sub-figure represents time in seconds and the y-axis represents stress levels in a range from 0 to 1.

## 5. Conclusions

In the current work, a dataset for continuous-valued stress level detection is presented. The dataset contains multimodal physiological and audio data from five participants following a stress-induction protocol that alternates between well-known stressors and relaxation moments. The physiological signals include ECG, RSP, and IMU signals and were acquired using a smart vest designed by us and equipped with all the aforementioned sensors. The audio data include recordings of the speech of the users throughout the whole duration of the experiments using a smartphone and a developed application.

In addition, a multimodal fusion solution based on audio and sensor data for accurate and real-time stress level detection for first responders is suggested. The solution is based on a decision-level fusion utilizing results from the physiological sensor and audio stress detection. Physiological sensor stress detection is based on extracting features from the ECG, RSP, and IMU signals, selecting the most relevant features utilizing a GA-based feature selection technique, and performing a regression analysis using an XGB machine learning algorithm. The audio analysis consists of segmenting the audio recording into smaller windows, followed by a feature extraction process, and finally the numeric estimation of the stress value using a Gaussian process regression model. The fusion of audio- and sensor-based stress levels is performed by utilizing an SVM regressor. Results on training data reveal that by fusing audio- and sensor-based stress levels, the MSE of the system is reduced from 0.0567 to 0.0062, thus improving the overall performance of the system. The whole system has also been tested in a real-life disaster management pilot scenario, where first responders were in the field equipped with the designed smart vest and an interface for recording their speech. The whole system operates in real-time and has reasonable results, as indicated by the pilot results.

**Author Contributions:** Conceptualization, V.-R.X. and M.D.; methodology, V.-R.X. and M.D.; data acquisition, F.Z. and M.M.; formal analysis, V.-R.X. and M.D.; investigation, V.-R.X., M.D. and A.T.; writing—original draft preparation, V.-R.X., M.D. and J.G.; writing—review and editing, S.S., S.D., L.W. and S.V.; supervision, L.W., S.V. and I.K. All authors have read and agreed to the published version of the manuscript.

**Funding:** This work was supported by the XR4DRAMA project funded by the European Commission (H2020) under the grant number 952133.

**Institutional Review Board Statement:** The study was conducted in accordance with the Declaration of Helsinki, and approved by the Ethics Advisory Board of the xR4DRAMA project.

**Informed Consent Statement:** Informed consent was obtained from all subjects involved in the study.

**Data Availability Statement:** The data that support the findings of this study are available from the corresponding author, V.R.X., upon reasonable request in the following repository (https://github.com/VasilisXeft/XR4DRAMA-Dataset-Stress, accessed on 27 April 2023).

**Conflicts of Interest:** These authors declare no conflict of interest.

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
