# Peer review of "A Multimodal Late Fusion Framework for Physiological Sensor and Audio-Signal-Based Stress Detection: An Experimental Study and Public Dataset"

_electronics, doi:10.3390/electronics12234871_

Round 1

Reviewer 1 Report

Comments and Suggestions for Authors

The research presents a very interesting topic in this paper, as well as results that are of wider significance when it comes to improving knowledge in the case of stress signals evaluation.  
However, some significant comments must be noted:
1.    The list of affiliations is incomplete. No address of the institution, etc. Look at the journal guidance.
2.    The title requires some editing – too many “and” words, which makes the title to sound not grammatically correct. “And” after  “physiological sensor” is not necessary.
3.    The abstract is clear and allows even not familiar readers to get a sense of the paper.
4.    In the case of the language, it is mostly good with some confusion in case of using an impersonal form of writing and persol (we, our etc.). Please unify.  Also, some grammar mistakes are visible.

5.    The introduction:
a.    A good introduction to the background of the paper is presented at the beginning.
b.    Later, some techniques are presented in connection with stress monitoring. However, most of them are presented like EEG, but at the same time, authors should mention and reference ECG (which is a technique used in the article). This can be done in connection with the importance of the topic and also in the case of fetal stress detection e.g. doi: 10.3934/mbe.2021250. Of course, the authors have dedicated chapters for related works, but this general view should be included just as one sentence in the introduction where some alternatives are mentioned.
6.    In some places, the article looks more like part of the thesis or the part of a report, e.g., an introduction to each section, e.g. “In this Section, the….”, than a scientific paper.
7.    The graphical output of the paper is very limited and, in some places, does not exist (e.g. chapter 3 would benefit from some graphical output, e.g. pictures). In other places, the figures are of very low quality, e.g. Fig.1 is very blurry.
This also must be improved for the result section- e.g. Fig.4 and 5 are difficult to read or interpret.
8.    Methods are mostly well described. However, some details on data acquisition parameters are missing, and some elements are described in a very general way.
9.    The conclusions- are acceptable.
10.    No ethics statements at the end of the article are included, although in this case, they are required.
11.    Please check if the article is not under the limit of the words for the journal. It looks like it is slightly under the limit.

Conclusions:
The article is interesting for the case type of paper. However, it is rushed, with multiple problems to be solved before eventual publication. If the paper is considerably improved, especially in the most crucial elements pointed out above, it can be considered for eventual acceptance.

Comments on the Quality of English Language

The title requires some editing – too many “and” words, which makes the title to sound not grammatically correct. “And” after  “physiological sensor” is not necessary.

Also, some minor grammar mistakes are visible.

Reviewer 2 Report

Comments and Suggestions for Authors

The manuscript introduces a novel publicly available multimodal dataset related to stress detection, derived from physiological sensors and audio signals. The authors conducted the data collection, implemented an analysis of the dataset, and evaluated it with four machine learning algorithms. They also tested their framework in a real-life pilot scenario. Overall, I found the manuscript well-structured and easy to follow. The data collection and analysis are well-presented. The publicly available dataset is potentially of great interest to the research community and could foster future research work in this domain.

A few comments/suggestions:

1. The number of participants is relatively small compared to other publicly available datasets. For example, the WESAD dataset contains data from 15 subjects, and the SWELL-KW dataset contains data from 25 participants. By contrast, the dataset introduced only contains data from 5 subjects. The limited number of participants could potentially lead to overfitting issues.

2. Line 247-248, “All participants signed etc.” => What is “etc”?

3. Line 324, “five different machine learning algorithms” => “four”?

4. Figure 1 has a lower resolution compared to other figures, causing inconsistency. I suggest replacing it with a higher-resolution version.

5. Could the authors elaborate more on the parameters of the four machine learning algorithms? For example, what is the number of k in kNN?

6. Figure 3 shows that the GA model has a notably higher MSE than other approaches. Could the authors offer any insights into the possible reasons? Have the authors tried any types of tuning to address that?

7. The subfigures in Figure 5 are relatively small and difficult to read. I suggest zooming in the subfigures to make them clearer to readers. 

8. I checked the GitHub link (Line 426) provided by the authors, but it gave me a 404 error. Could the authors verify that this is the correct link?

Comments on the Quality of English Language

N/A

Round 2

Reviewer 1 Report

Comments and Suggestions for Authors

Dear Authors,

The submitted manuscript has been improved in accordance with the reviewer's suggestions. No other methodological problems were noticed.

Although elements can still be improved, they are primarily of editing type and can be improved at the final stage. This relates mostly to the figure's editing. 

Thus, the reviewer has no other request.

Best regards,

The reviewer.

Author Response

We would like to thank the reviewer for the comments and the time and effort spent in reviewing our paper.

Reviewer 2 Report

Comments and Suggestions for Authors

I'm grateful for the authors' significant efforts in improving the manuscript. The authors have addressed almost all my comments. I only have one minor comment related to the authors' response to Question 3.

The authors mentioned that they used five machine learning algorithms (Line 336). But at Line 295, they said "...four different machine learning algorithms...". I suggest making the numbers consistent.

Comments on the Quality of English Language

N/A

Author Response

We would like to thank the reviewer for his comments. Nevertheless, there is no inconsistency between the two numbers since they refer to different sets of machine learning algorithms. In line 295 we refer to the set of machine learning algorithms used for physiological signals analysis, which is a set of four different algorithms. On the other hand, in line 336 we refer to the set that was used for the fusion of physiological signals and audio data analysis results, which is a set of five different algorithms. Therefore both numbers are correct.